# Effects of organic coating on the nitrate formation by suppressing the $N_2O_5$ heterogeneous hydrolysis: A case study during wintertime in Beijing-Tianjin-Hebei (BTH)

Lang Liu[1,4], Jiarui Wu[1], Suixin Liu[1], Xia Li[1], Jiamao Zhou[1], Tian Feng[1], Yang Qian[3], Junji Cao[1,2], Xuexi Tie[1], Guohui Li[1,2*]

[1]Key Lab of Aerosol Chemistry and Physics, SKLLQG, Institute of Earth Environment, Chinese Academy of Sciences, Xi'an, China
[2]CAS Center for Excellence in Quaternary Science and Global Change, Xi'an, China
[3]State Key Laboratory of Environmental Criteria and Risk Assessment & Environmental Standards Institute, Chinese Research Academy of Environmental Sciences, Beijing, China
[4]University of Chinese Academy of Sciences, Beijing, China

*Correspondence to*: Guohui Li (ligh@ieecas.cn)

**Abstract:** Although stringent emission mitigation strategies have been carried out since 2013 in Beijing-Tianjin-Hebei (BTH), China, heavy haze with high levels of fine particulate matter (PM$_{2.5}$) still frequently engulfs the region during wintertime and the nitrate contribution to PM$_{2.5}$ mass has progressively increased. The $N_2O_5$ heterogeneous hydrolysis is the most important pathway of the nitrate formation at nighttime. In the present study, the WRF-Chem model is applied to simulate a heavy haze episode from 10 to 27 February 2014 in BTH to evaluate contributions of the $N_2O_5$ heterogeneous hydrolysis to nitrate formation and effects of organic coating. The model generally performs reasonably well in simulating meteorological parameters, air pollutants and aerosol species against observations in BTH. The $N_2O_5$ heterogeneous hydrolysis with all the secondary organic aerosol assumed to be involved in coating considerably improves the nitrate simulations compared to the measurements in Beijing. On average, organic coating decreases nitrate concentrations by 8.4% in BTH during the episode, and the $N_2O_5$ heterogeneous hydrolysis with organic coating contributes about 30.1% of nitrate concentrations. Additionally, the reaction also plays a considerable role in the heavy haze formation, with a PM$_{2.5}$ contribution of about 11.6% in BTH. Sensitivity studies also reveal that future studies need to be conducted to predict the organic aerosol hygroscopicity for accurately representing the organic coating effect on the $N_2O_5$ heterogeneous hydrolysis.

## 1 Introduction

Within recent decades, China has been suffering from pervasive and persistent haze pollution caused by elevated levels of fine particulate matters ($PM_{2.5}$), particularly in Beijing-Tianjin-Hebei (BTH) ( Guo et al., 2014; Gao et al., 2016; Wang et al., 2016). Numerous studies have revealed that the inorganic aerosols, including nitrate, sulfate and ammonium, are the most abundant component of $PM_{2.5}$ during haze pollution episodes in BTH, and that the evolution of the haze pollution is characterized by the formation of substantial amounts of sulfate and nitrate (Sun et al., 2013; Zhang et al., 2013; Zhao et al., 2013; Sun et al., 2015). Since 2013, several aggressive emission control strategies have been implemented in China, including desulfurization and dedusting for coal combustion, vehicle restriction and executing stringent emission standards in the key industries (Tao et al., 2017). However, the control of emissions of nitrate gaseous precursors does not seem to be effective, since many observations have shown that the nitrate aerosol concentration has progressively increased in recent several years (Zhang et al., 2012; Sun et al., 2015; Zhang et al., 2015; Tao et al., 2017).

In the atmosphere, nitrate aerosol is formed via nitrous acid ($HNO_3$) to balance the inorganic cations in the aerosol phase. $HNO_3$ is produced through four pathways (Kim et al., 2014): (1) the reaction of OH and $NO_2$ (main gas phase pathway and usually considered as the daytime pathway because the OH radical is severely limited at night due to lack of $O_3$ and peroxide photolysis), (2) $NO_3$ radical reaction with hydrocarbons, (3) aqueous reaction of $NO_3$ radical to form $HNO_3$, and (4) $NO_3$ conversion to $N_2O_5$ with subsequently heterogeneous chemical conversion to form $HNO_3$. The last pathway is referred to as the most important pathway during nighttime, since both $NO_3$ and $N_2O_5$ are photolytically liable, or even under heavy haze situation with weak sunlight and high relative humidity (RH) (Brown et al., 2016).

The heterogeneous hydrolysis of $N_2O_5$ on the surface of deliquescent aerosols to form
$HNO_3$ is quantified by the reaction probability ($\gamma_{N_2O_5}$) (Bertram and Thornton, 2009; Chen et
al., 2018; Davis et al., 2008; Riemer et al., 2003). $\gamma_{N_2O_5}$ has been measured by previous
laboratory experiments, dependent on particulate chemical composition, RH, temperature,
aerosol surface area and water content (Chang et al., 2011), and in the order of $10^{-2}$ (Zheng et
al., 2015). In modeling studies, various parameterizations of $\gamma_{N_2O_5}$ are used to simulate the
nitrate formation. Dentener and Crutzen (1993) have first used 0.1 as the representative
$\gamma_{N_2O_5}$ in a three-dimensional global model. Riemer et al. (2003) have developed a $\gamma_{N_2O_5}$
parameterization on the surface of aerosols containing sulfate and nitrate (hereafter referred
as to Riemer03), which has widely been used and further improved in air quality models.
Davis et al. (2008) have implemented a $\gamma_{N_2O_5}$ parameterization on the surface of particles
containing sulfate, nitrate, ammonium, as a function of RH, temperature and phase state, to
improve simulations of $N_2O_5$ hydrolysis. Bertram and Thornton (2009) have developed a
parameterization to consider the influence of chloride salts on $\gamma_{N_2O_5}$ as a function of RH.
The coating of particles by organic materials has been reported to inhibit $N_2O_5$ uptake
(Anttila et al., 2006), and suggested as a possible explanation for field observations of
suppressed $N_2O_5$ uptake (Brown et al., 2006). Evans and Jacob (2005) have incorporated a
$\gamma_{N_2O_5}$ parameterization on surfaces of sulfate particles as a function of RH and temperature
into the GEOS-CHEM model, including the effects of dust, sea salt, sulfate, elemental carbon
and organic carbon but ignoring the nitrogen-containing species. Riemer et al. (2009) have
developed a $N_2O_5$ uptake parameterization (Riemer09) based on the laboratory results of
Anttila et al. (2006), which combines the nitrogen-containing and organic effects on $N_2O_5$
hydrolysis. The parameterization has been used to estimate the maximum effect of organic
coating by assuming that all available secondary organic compounds (SOC) contribute to the
coating in a 3-D model. The results show that SOC could suppress $N_2O_5$ uptake significantly,
reducing particulate nitrate concentrations by up to 90%. Lowe et al. (2015) have further
combined the organic coating and chloride salts effects on $\gamma_{N_2O_5}$ in the WRF-Chem model.
Most recently, Chen et al. (2018) have developed a new $\gamma_{N_2O_5}$ parameterization with respect
to RH, temperature, and aerosol composition, showing that organic coating effect on $\gamma_{N_2O_5}$
is not as important as expected over western and central Europe. However, there is still a lack
of modeling studies focused on the effect of organic coating on $\gamma_{N_2O_5}$ and particulate nitrate
formation in China. Wang et al. (2017) have evaluated the potential particulate nitrate
formation through the $N_2O_5$ hydrolysis reaction without considering the organic coating
effect during a haze pollution episode in Beijing, and found that the observed nitrate
concentration (20.6 $\mu g\ m^{-3}$ on average) is lower than the assessment (57.0 $\mu g\ m^{-3}$ on average).
Considering the high organic aerosol concentration and increasing trend of particulate nitrate
during haze days in BTH, it is imperative to assess the effect of organic coating on $N_2O_5$
hydrolysis and its consequent contribution to the nitrate formation.
In the present study, based on Riemer09 parameterization, the contribution of the
organic coating effect on $N_2O_5$ hydrolysis to the nitrate formation is investigated using the
WRF-Chem model. The model configuration and methodology are described in Section 2.
Results and sensitivity studies are presented in Section 3. Discussion and summary are given
in section 4.

**2    Model and Methodology**
**2.1 WRF-Chem model and configuration**
A modified version of the WRF-Chem model (Grell et al., 2005) is used in this study,
which is developed by Li et al. (2010; 2011a; 2011b; 2012) at the Molina Center for Energy
and the Environment. A new flexible gas phase chemical module has been developed and
implemented into the version of the WRF-Chem model, which can be utilized with different

chemical mechanisms, including CBIV, RADM2, and SAPRC. The gas-phase chemistry is solved by an Eulerian backward Gauss-Seidel iterative technique with a number of iterations, inherited from NCAR-HANK (Hess et al., 2000). In the study, the SAPRC99 chemical mechanism is used based on the available emission inventory. For the aerosol simulations, the CMAQ/models3 aerosol module (AERO5) developed by US EPA has incorporated into the model (Binkowski and Roselle, 2003). Briefly, the wet deposition uses the method in the CMAQ module and the dry deposition of chemical species is parameterized following Wesely (1989). The photolysis rates are calculated using the Fast Tropospheric Ultraviolet and Visible Radiation Model (FTUV; Tie et al., 2003; Li et al., 2005), with the aerosol and cloud effects on the photochemistry (Li et al., 2011a).

ISORROPIA (version 1.7) is used to predict the thermodynamic equilibrium between the ammonia-sulfate-nitrate-chloride-water aerosols and their gas phase precursors of $H_2SO_4$-$HNO_3$-$NH_3$-HCl-water vapor. It is worth noting that the most recent extension of ISORROPIA, known as ISORROPIA II, has incorporated a larger number aerosol species (Ca, Mn, K salts) and is designed to be a superset of ISORROPIA (Fountoukis and Nenes, 2007). Considering that crustal species are not considered in the study, ISORROPIA (version 1.7) is still used to calculate inorganic components and ISORROPIA II is imperative to be incorporated into the WRF-Chem model in future studies. In addition, a parameterization of sulfate heterogeneous formation involving aerosol liquid water (ALW) has been developed and implemented into the model, which has successfully reproduced the observed rapid sulfate formation during haze days (Li et al., 2017). The sulfate heterogeneous formation from $SO_2$ is parameterized as a first order irreversible uptake by ALW surfaces, with a reactive uptake coefficient of $0.5\times10^{-4}$ assuming that there is enough alkalinity to maintain the high iron-catalyzed reaction rate.

The OA module is based on the VBS approach with aging and detailed information can

be found in Li et al. (2011b). The POA components from traffic-related combustion and
biomass burning are represented by nine surrogate species with saturation concentrations (C*)
ranging from $10^{-2}$ to $10^{6}$ μg m$^{-3}$ at room temperature (Shrivastava et al., 2008), and assumed
to be semi-volatile and photochemically reactive (Robinson et al., 2007). The SOA formation
from each anthropogenic or biogenic precursor is calculated using four semi-volatile VOCs
with effective saturation concentrations of 1, 10, 100, and 1000 μg m$^{-3}$ at 298 K. The SOA
formation via the heterogeneous reaction of glyoxal and methylglyoxal is parameterized as a
first-order irreversible uptake by aerosol particles with an uptake coefficient of $3.7 \times 10^{-3}$
(Liggio et al., 2005; Zhao et al., 2006; Volkamer et al., 2007). The OA module has reasonably
reproduced the POA and SOA concentration against measurements, and detailed model
performance can be found in Li et al. (2011b), Feng et al. (2016), and Xing et al. (2019).

The anthropogenic emission inventory with a horizontal resolution of 6 km is developed

by Zhang et al. (2009), with the base year of 2013, including industry, transportation, power
plant, residential and agriculture sources. The Model of Emissions of Gases and Aerosols
from Nature (MEGAN) is used to calculate the biogenic emissions online (Guenther et al.,

2006).

A heavy haze episode from 10 to 27 February 2014 in BTH is simulated in association

with the field observation of air pollutants and secondary inorganic aerosols. Detailed model
configuration can be found in Table 1 and the simulation domain is presented in Figure 1.
**2.2 Parameterization of the heterogeneous hydrolysis of $N_2O_5$**

The reaction of $N_2O_5$ heterogeneous hydrolysis on the surface of deliquescent aerosols

to form $HNO_3$ can be represented as:
$N_2O_5 + H_2O \rightarrow 2 \cdot HNO_3$            (Eq. 1)
This reaction is usually implemented into air quality transport models as a first order loss:
$\frac{\partial [N_2O_5]}{\partial t} = -k_{N_2O_5} \cdot [N_2O_5]$            (Eq. 2)
$[N_2O_5]$ represents the N₂O₅ concentration in the atmosphere. The loss rate constant, $k_{N_2O_5}$, is
parameterized in the following way:
$$k_{N_2O_5} = \frac{1}{4} \cdot c_{N_2O_5} \cdot S \cdot \gamma_{N_2O_5}$$      (Eq. 3)
Where $c_{N_2O_5}$ is the average molecular velocity of N₂O₅, and S is the available aerosol
surface area density.
In this study, the parameterization of $\gamma_{N_2O_5}$ follows Riemer03 and Riemer09. In the
parameterization, the primary emission compounds such as element carbon, insoluble organic
matter (most part of POA), insoluble inorganic and mineral dust particles are assumed to
serve as a nucleus of aerosols. Condensation of soluble chemical components and further
water vapor on the surface of the nucleus forms an aqueous layer. The nucleus and the
aqueous layer are assumed as unified "core" (aqueous core) in Riemer03 and Riemer09
parameterizations. In Riemer03 parameterization, soluble inorganic components including
sulfate and nitrate are taken into consideration for suppressing the N₂O₅ heterogeneous
hydrolysis uptake in the aqueous core, and the parameterization of $\gamma_{N_2O_5}$ is defined as:
$$\gamma_{N_2O_5} = f \cdot \gamma_1 + (1-f) \cdot \gamma_2$$      (Eq. 4)
with $\gamma_1 = 0.02$ and $\gamma_2 = 0.002$, and $f$ is defined as:
$$f = \frac{m_{SO_4^{2-}}}{m_{SO_4^{2-}} + m_{NO_3^-}}$$      (Eq. 5)
$m_{SO_4^{2-}}$ and $m_{NO_3^-}$ are the aerosol mass concentrations of soluble sulfate and nitrate.
In Riemer09 parameterization, unreactive organic layers are further considered for the
suppression of N₂O₅ hydrolysis by covering the aqueous core. Organic layers may be formed
by secondary organic aerosols, and such layers may consist of a single layer of molecules
(monolayered coatings) or of several molecule layers (multilayered coatings) on the surface
of the aqueous core. These organic layers are assumed as organic "coating" (shell) in the
Riemer09 parameterization. The resistor scheme to calculate $\gamma_{N_2O_5}$ in Riemer09
parameterization is parameterized as follows:
$$\frac{1}{\gamma_{N_2O_5}} = \frac{1}{\gamma_{N_2O_5,core}} + \frac{1}{\gamma_{N_2O_5,coat}}$$    (Eq. 6)
where $\gamma_{N_2O_5,core}$ is the reaction probability of the aqueous core which is calculated using Eq.
4, and $\gamma_{N_2O_5,coat}$ is the pseudo-reaction probability of the organic coating calculated by the
following formulation:
$$\gamma_{N_2O_5,coat} = \frac{4 \cdot R \cdot T \cdot H_{org} \cdot D_{org} \cdot R_c}{c_{N_2O_5} \cdot \ell \cdot R_p}$$    (Eq. 7)
Where $R$ is the universal gas constant, $T$ is temperature, $H_{org}$ is the Henry's law constant for
$N_2O_5$ in the organic coating, and $D_{org}$ is the diffusion coefficient for $N_2O_5$ in the organic
coating. $H_{org}$ and $D_{org}$ depend on the physicochemical properties of the compounds
comprising the organic coating. In the Riemer09 scheme, $H_{org} \cdot D_{org}$ is defined as
$0.03 \cdot H_{aq} \cdot D_{aq}$. $H_{aq}$ is the Henry's law constant of $N_2O_5$ for the aqueous phase ($H_{aq}$ = 5000
M atm$^{-1}$) and $D_{aq}$ is the diffusion coefficient of $N_2O_5$ in the aqueous phase ($D_{aq}$ = $10^{-9}$ m$^2$
s$^{-1}$). $R_p$, $R_c$, and $\ell$ are the radius of the particle, radius of the inorganic core, and thickness
of the coating, respectively. $R_p$, $R_c$, and $\ell$ are calculated as follows:
$$R_p = R_c + \ell$$    (Eq. 8)
$$\ell = R_p \cdot (1 - \beta^{\frac{1}{3}})$$    (Eq. 9)
$$\beta = \frac{V_{inorg}}{V_{inorg} + V_{org}}$$    (Eq. 10)
Where $V_{inorg}$ and $V_{org}$ are the volume of inorganic and organic materials, respectively.
**2.3 Statistical methods for model evaluation**
In this study, the mean bias (MB), root mean square error (RMSE), the index of
agreement (IOA), mean fractional bias (MFB) and mean fractional error (MFE) are used to
evaluate the model performance in simulating air pollutants.
$$MB = \frac{1}{N}\sum_{i=1}^{N}(P_i - O_i)$$    (Eq. 11)
$$RMSE = \left[ \frac{1}{N} \sum_{i=1}^{N} (P_i - O_i)^2 \right]^{\frac{1}{2}}$$ (Eq. 12)
$$IOA = 1 - \frac{\sum_{i=1}^{N}(P_i - O_i)^2}{\sum_{i=1}^{N}(|P_i - \bar{O}| + |O_i - \bar{O}|)^2}$$ (Eq. 13)
$$MFB = \frac{1}{N} \sum_{i=1}^{N} \frac{P_i - O_i}{(P_i + O_i)/2}$$ (Eq. 14)
$$MFE = \frac{1}{N} \sum_{i=1}^{N} \frac{|P_i - O_i|}{(P_i + O_i)/2}$$ (Eq. 15)
Where $P_i$ and $O_i$ are the simulated and observed variables, respectively. N is the total
number of the simulations for comparisons, and $\bar{O}$ donates the average of the observation.
The IOA ranges from 0 to 1, with 1 showing a perfect agreement of the simulation with the
observation.

## 2.4 Air pollutants observations

Simulations are compared to available meteorological and air pollutants observations to
evaluate the model performance. The meteorological parameters including temperature, RH,
wind speed and direction are obtained from the website http://www.meteomanz.com. The
hourly observations of $PM_{2.5}$, $O_3$, $SO_2$, $NO_2$, and CO concentrations are released by China
National Environmental Monitoring Center and can be downloaded from the website
http://106.37.208.233:20035.
Additionally, hourly OC and EC concentrations are measured using a thermal/optical
reflectance carbon analyzer (OCEC RT-4, Sunset Lab, USA) at Chinese Research Academy
of Environmental Sciences (CRAES, 40.04°N, 116.40°E) in Beijing (Wei et al., 2014; Liu et
al., 2018). Hourly sulfate, nitrate, ammonium, and other inorganic ions are sampled and
analyzed by ion chromatography (URG 9000S, Thermo Fisher Scientific, USA).
The OC/EC ratio approach is used to derive the SOA mass concentrations from EC and
OC filter measurements as follows (Strader, 1999; Cao et al., 2004):
$$POC = EC \times \left( \frac{POC}{EC} \right)$$ (Eq. 16)
$$SOC = OC - POC \qquad\qquad\qquad\text{(Eq. 17)}$$
$$SOA = SOC \times \left(\frac{SOA}{SOC}\right) \qquad\qquad\qquad\text{(Eq. 18)}$$
Where POC and SOC are the primary OC and secondary OC, respectively. In the present
study, $\frac{POC}{EC}$ and $\frac{SOA}{SOC}$ are assumed to be 2.4 and 1.6, respectively, based on the previous
studies (Cao et al., 2007; Aiken et al., 2008; Yu et al., 2009) and detailed information about
the approach can be found in Feng et al. (2016). It is worth noting that those assumed $\frac{POC}{EC}$
and $\frac{SOA}{SOC}$ could potentially affect the model-measurement comparisons.

**3    Results and discussion**
**3.1 Synoptic conditions during the wintertime of 2014**
Based on the NCEP FNL reanalysis data (https://rda.ucar.edu/datasets/ds083.2), we have
initially performed the analysis of synoptic conditions using the wind, temperature, relative
humidity, and geopotential height fields at 500hPa and 850hPa averaged from 10 to 27
February 2014 over China, respectively (Figure 2). At 500hPa, flat westerly winds prevail
over BTH and its surrounding area, indicating a stagnant atmospheric circulation conditions
(Figure 2a). Moreover, the flat isotherm distribution is similar to that of the isobar at 500hpa,
showing that there is no obvious exchange of clod and warm air masses, which together with
the flat westerly leads to the weak turbulent mixing in the vertical direction and the stable
weather condition (Figure 2b). At 850hPa, the southeast coastal areas of China are controlled
by the anti-cyclone whose center locates over the South China Sea (Figure 2c). In the eastern
China influenced by the anti-cyclone, the weak southerly wind prevails over the BTH and its
surrounding regions, providing a favorable condition for stagnant weather conditions and
further the formation of air pollution. With the prevailing southerly wind, the warm and
humid air flow and the polluted air mass are subject to being transported from south to north,
aggravating the air pollution in BTH. In addition, high relative humidity conditions facilitate
the heterogeneous reactions for the secondary aerosol formation (Figure 2d).
**3.2  Model performance**
In order to quantify effects of the $N_2O_5$ heterogeneous hydrolysis and organic coating on
the nitrate formation, three experiments have been performed in the study. In the base case,
Riemer09 parameterization is used to take into consideration the organic coating effect on the
$N_2O_5$ heterogeneous hydrolysis by assuming that all the SOA is involved in coating (hereafter
referred to as B-case). In the first sensitivity case, the contribution of $N_2O_5$ heterogeneous
hydrolysis to the nitrate formation is not considered (hereafter referred to as H0-case); In the
second sensitivity case, the organic coating effect is not considered in Riemer09
parameterization (hereafter referred to as C0-case). The simulation results in the B-case are
compared to observations in BTH.
3.2.1 Meteorological parameters simulations in Beijing
Considering that the meteorological conditions play a crucial role in air pollution
simulations, which determine accumulation or dispersion of pollutants, verifications are first
performed for the simulations of meteorological fields. Figure 3 presents the temporal profile
of the simulated and observed temperature, RH, wind speed, and wind direction averaged
over 12 meteorological sites in Beijing from 10 to 27 February 2014. The WRF-Chem model
reproduces well the temporal variation of the surface temperature during the whole episode.
The MB and RMSE is -0.2 and 1.7°C, and the IOA reaches 0.94, indicating good agreement
of the simulations with observations (Table 2). The simulated temporal RH variations are also
well consistent with observations, with the MB, RMSE and IOA of 2.6%, 10.9% and 0.89,
respectively. In addition, the model reasonably well tracks the temporal variations of the
surface wind, with IOAs of 0.73 and 0.66 for the wind speed and direction, respectively.
3.2.2 Air pollutants simulations in BTH

Figure 4 shows the relationship between observed and simulated mass concentrations of

$PM_{2.5}$, $O_3$, $SO_2$, $NO_2$, and CO in Beijing, Tianjin, and Hebei from 10 to 27 February 2014.
The correlation coefficient (R) of $PM_{2.5}$ mass concentrations between observations and
simulations in Beijing, Tianjin, and Hebei is 0.83, 0.80, and 0.90, respectively, indicating a
good performance of the WRF-Chem model in simulating the $PM_{2.5}$ concentration in BTH.
The correlation of $O_3$ and $NO_2$ mass concentrations between observations and simulations is
not as good as that of $PM_{2.5}$ concentrations in BTH, with the R between 0.6 and 0.8.
Apparently, the R of $SO_2$ simulations with observations show that the WRF-Chem model still
has difficulties in well simulating $SO_2$ concentrations in BTH, particularly in Hebei. Except
uncertainties from $SO_2$ emissions, such as source intensities and distributions, diurnal profiles,
et al., the bias of simulated wind fields also substantially influences the $SO_2$ simulation (Bei
et al., 2017). Particularly, $SO_2$ is principally emitted by the point source, including the power
plants and agglomerated industrial zones, so the $SO_2$ simulations is more sensitive to the
wind field simulation uncertainties. In terms of R, the $SO_2$ simulations in Beijing and Tianjin
is better than those in Hebei, indicating that the $SO_2$ emissions in Beijing and Tianjin are
generally determined by area sources, i.e., the residential living, but the point source
dominates the $SO_2$ concentration in Hebei. Considering the long-life time of CO in the
atmosphere, the CO simulation is decided by its emission and the meteorological fields. The
R of CO simulations with observations in BTH ranges from around 0.6 to 0.7, showing that
the CO emissions used in the study and simulated meteorological fields are generally
reasonable.

Figure 5 presents the diurnal profiles of simulated and observed $PM_{2.5}$, $O_3$, $NO_2$, $SO_2$,

and CO mass concentrations averaged over all ambient monitoring stations in BTH during
the simulated episode. The WRF-Chem model well reproduces the diurnal variations of the
$PM_{2.5}$ mass concentrations against observations in BTH. The MB and RMSE is -6.3 and
27.6μg m$^{-3}$, respectively, and the IOA is 0.96. The model generally well replicates the haze
developing stage, but fails to capture the observed spikes of PM$_{2.5}$ mass concentrations,
which might be caused by the uncertainty of the simulated meteorological fields or irregular
air pollutants emissions (Bei et al., 2017). The simulated O$_3$ diurnal variations are in good
agreement with observations, with the MB and IOA of 1.4 μg m$^{-3}$ and 0.91, respectively. The
model tracks well the observed diurnal variations of NO$_2$ mass concentrations with an IOA of
0.92, but it slightly overestimates NO$_2$ concentrations compared to observations with a MB of
6.6 μg m$^{-3}$. However, during nighttime, the model overestimation is considerable, which is
perhaps due to the model biases in modeling nighttime PBL. Although the model reasonably
yields the variation trend of the observed SO$_2$ concentration, with an IOA of 0.85, the
dispersion of the simulated SO$_2$ concentration is rather large, with a RMSE of 27.8 μg m$^{-3}$. In
addition, the model overestimates the SO$_2$ concentration compared to observations, and the
MB is 7.6 μg m$^{-3}$, which might be mainly caused by the emission inventory that has
undergone noticeable changes since implementation of emission control strategies in 2013 in
BTH. The model performs well in simulating CO diurnal variations against observations,
with the MB and IOA of 0.2 μg m$^{-3}$ and 0.90, respectively.

Figure 6 presents the distributions of simulated and observed near-surface mass

concentrations of PM$_{2.5}$, O$_3$, NO$_2$, and SO$_2$ along with the predicted wind fields averaged
during the episode. Generally, the simulated wind in BTH is weak during the episode and the
southerly wind prevails, well corresponding to the synoptic situation at 850hPa and 500hPa,
which is favorable for the accumulation of air pollutants. The observed PM$_{2.5}$ concentrations
are more than 115 μg m$^{-3}$ on average, showing that BTH suffers from heavy haze pollution
(Figure 6a). The model generally well reproduces the spatial distribution of PM$_{2.5}$
concentrations against observations, with the PM$_{2.5}$ concentration exceeding 150 μg m$^{-3}$ in
the plain area of BTH. The simulated and observed O$_3$ mass concentrations are less than 50
$\mu g\ m^{-3}$ in the plain area of BTH, and in several megacities, including Beijing, Tianjin,
Baoding, and Shijiazhuang, the $O_3$ concentrations are less than 30 $\mu g\ m^{-3}$ (Figure 6b). The
low $O_3$ concentrations during the episode are generally caused by the weak insolation during
wintertime, which is unfavorable for photochemical reactions, and the titration due to high
$NO_x$ emissions in BTH (Figure 6c). The simulated $NO_2$ concentrations are generally more
than 40 $\mu g\ m^{-3}$, consistent with the observations at monitoring sites in BTH. The simulated
and observed $SO_2$ mass concentrations in cities or their surrounding areas are still rather high,
exceeding 50 $\mu g\ m^{-3}$ (Figure 6d). Elevated $SO_2$ concentrations in BTH during wintertime are
to some degree contributed by the residential coal combustion (Li et al., 2018). High levels of
$NO_2$ and $SO_2$ show that stringent emission mitigation strategies still need to be implemented
in BTH.
3.2.3 Sulfate, ammonium and SOA simulations in Beijing
The SOA and sulfate concentration directly influences the $N_2O_5$ heterogeneous
hydrolysis in the Riemer09 parameterization, and the ammonium aerosol concentration
substantially affects the nitrate aerosol formation. Therefore, Figure 7 presents the temporal
profiles of observed and calculated SOA, sulfate, and ammonium mass concentrations at
CRAES site in Beijing 10 to 27 February 2014. The model reasonably tracks the diurnal
variation of the SOA concentration compared to observations, with the MB and IOA of -1.2
$\mu g\ m^{-3}$ and 0.83, respectively. The observed SOA concentration exhibits rather large
fluctuations, which is not well reproduced by the model. The simulated sulfate trend is
generally in agreement with observations with an IOA of 0.88, but there are considerable
model biases. During the first pollution event, the model reasonably reproduces the sulfate
increase during the haze developing stage, but the early falloff of sulfate concentrations
during the dissipation stage causes the substantial underestimation. However, during the
second pollution event, the model considerably overestimates the sulfate concentration
against the measurement from 22 to 26 February 2014. The ammonium simulation is slightly
better than that of sulfate, with an IOA of 0.90.
Furthermore, the MFB and MFE between simulations and observations are also
calculated to evaluate the model performance in simulating meteorological parameters and air
pollutants (Table 2). Boylan and Russell (2006) have proposed that MFB should be within
±60% and MFE should be below 75% for a satisfactory model performance. For the
simulation in the B-case, MFB values are within 27% and MFE values are below 55%,
indicating that the model performance is satisfactory.
In summary, the WRF-Chem model performs reasonably well in simulating air pollutants
and aerosol species, showing that the simulated meteorological fields and emissions used in
the study are generally reasonable.
**3.3 Contributions of the $N_2O_5$ heterogeneous hydrolysis and organic coating to the**
**nitrate formation**
Figure 8 provides the nitrate temporal variations in the three cases against observations
at CRAES site in Beijing from 10 to 27 February 2014. When the $N_2O_5$ heterogeneous
hydrolysis is not considered in the H0-case, although the model well tracks the observed
nitrate variations with an IOA of 0.91, it considerably underestimates nitrate concentration
against the measurement, with a MB of -17.0 μg m$^{-3}$. When the $N_2O_5$ heterogeneous
hydrolysis is taken into consideration based on the Riemer09 parameterization without
organic coating in the C0-case, the nitrate simulation is improved compared to that in the
H0-case, with an IOA of 0.95. However, the model commences to overestimate the nitrate
concentration compared to the measurement, with a MB of 5.4 μg m$^{-3}$. In the B-case, when
all the SOA is assumed to be involved in coating to suppress the $N_2O_5$ heterogeneous uptake
on surfaces of deliquescent aerosols, the model performs best in simulating the nitrate
variation compared to the measurement, with the MB and IOA of 0.1 μg m$^{-3}$ and 0.96,
respectively. The remarkable consistency of the simulated nitrate in the B-case with the
measurement indicates that the organic coating plays an important role in improving the
nitrate simulation. It is worth noting that the MB for nitrate aerosols at CRAES site in the
B-case is close to zero, but the RMSE is still rather large, reaching 19.0 $\mu g\ m^{-3}$, showing
considerable underestimation and overestimation, caused by uncertainties of meteorological
fiends and emissions. For example, the model overestimates nitrate concentrations on 11, 13,
and 14 February and underestimation on 24 February against measurements. In addition, the
early occurrence of intensified winds in the morning on 16 February in simulations cause
rapid falloff of nitrate concentrations, leading to substantial model biases.

Figure 9a presents the distribution of contributions of the $N_2O_5$ heterogeneous

hydrolysis to the nitrate formation averaged during the episode by differentiating simulations
in the B-case and H0-case. The contribution of the $N_2O_5$ heterogeneous hydrolysis to the
nitrate formation is substantial in BTH, exceeding 15 $\mu g\ m^{-3}$ in the plain area. Although the
$O_3$ concentration is fairly low in BTH during the episode (Figure 6b), particularly during
nighttime (Figure 5b), the elevated $NO_2$ level still facilitates the $N_2O_5$ formation to warrant
occurrence of the $N_2O_5$ heterogeneous hydrolysis. Previous studies have revealed that $N_2O_5$
heterogeneous hydrolysis is vital in nitrate formation. For example, Wang et al. (2017) have
calculated the daily average nitrate formation potential from the $N_2O_5$ heterogeneous
hydrolysis in Beijing, showing that the reaction accounts for 52% of the total nitrate
formation. Su et al. (2017) have investigated the contribution of $N_2O_5$ heterogeneous
hydrolysis to the nitrate formation in Beijing during autumn in 2015 and found that the
reaction causes a 21.0% enhancement of nitrate concentrations. In the present study, the
nitrate contribution of the $N_2O_5$ heterogeneous hydrolysis is 29.4% in Beijing during the
episode on average, which is close to the result in Su et al. (2017) but much lower than that in
Wang et al. (2017). The average nitrate contribution of the reaction in BTH is about 30.1%,
showing that the reaction constitutes an important nitrate source during the haze pollution
episode. Additionally, the $N_2O_5$ heterogeneous hydrolysis contributes 11.6% of the $PM_{2.5}$
concentration on average, playing a considerable role in the haze formation in BTH.

However, it is worth noting that the brute force method (BFM) is used to quantify the

contribution of the $N_2O_5$ heterogeneous hydrolysis to the nitrate formation (Dunker et al.,
1996). The BFM is generally used to assess the importance of some source, but it lacks
consideration of interactions of the complicated physical and chemical processes in the
atmosphere (Zhang and Ying, 2011). Therefore, in the study, the contribution of the $N_2O_5$
heterogeneous hydrolysis to the nitrate formation might be underestimated, considering the
competition of inorganic cations from $HNO_3$ formed through gas-phase reactions and sulfate
aerosols in the atmosphere. It is imperative to use the source-oriented base module to
evaluate the nitrate contribution of the reaction.

Figure 9b shows the distribution of the average decrease of nitrate concentrations due to

suppression of organic coating during the episode by differentiating simulations in the B-case
and C0-case. The organic coating reduces the nitrate concentration by more than 5 μg m$^{-3}$ in
the plain area of BTH, and on average, the decrease of nitrate aerosols is 4.7 μg m$^{-3}$ or 8.4%
in BTH during the episode. Riemer et al. (2009) have shown that when the nitrate levels are
high (above 15 μg m$^{-3}$), the organic coating decreases nitrate concentrations by 10-15% over
Europe. However, Cheng et al. (2018) have demonstrated that the suppression of organic
coating is negligible over western and central Europe, with an influence on nitrate
concentrations of less than 2% on average and 20% at the most significant moment.
Apparently, except $N_2O_5$ and water soluble OA in the atmosphere, the effect of organic
coating is also dependent on $NH_3$, RH, and temperature. Hence, the inconsistency between
the model results about the organic coating effect can be attributed to the variation in
simulation conditions. For example, in order to obtain substantial effects of the organic
coating, $N_2O_5$, SOA, and $NH_3$ need to be present when RH is high and temperature is low.
However, those conditions are rarely fulfilled simultaneously over western and central
Europe, causing a negligible effect of organic coating (Chen et al., 2018). Additionally, Wang
et al. (2017) have indicated that the evaluated nitrate level with the $N_2O_5$ heterogeneous
hydrolysis in Beijing is much higher than the observation, which they have attributed to
atmospheric dilution and deposition. It is worth noting that the organic coating effect might
constitute one of the most possible reasons for the overestimation of nitrate concentrations,
considering the elevated SOA level in Beijing which suppresses the $N_2O_5$ heterogeneous
hydrolysis and results in high observed $N_2O_5$ concentrations (Wu et al., 2017). Figure 10
presents the temporal variation of the simulated $\gamma_{N_2O_5}$ in Beijing during the episode. The
simulated $\gamma_{N_2O_5}$ fluctuates between 0.009 and 0.02 when organic coating is included, with
an average of 0.013. The estimated $\gamma_{N_2O_5}$ in Beijing by Wang et al. (2017) ranges from
0.025 to 0.072 without consideration of the suppression of organic coating, indicating that
organic coating substantially hinders the $N_2O_5$ heterogeneous hydrolysis, likely causing the
observed high level of $N_2O_5$ during nighttime.

It is worth noting that, in the study, the assumption of metastable aerosols is used or the

water soluble aerosol is assumed to be only in liquid state in simulations. However, Wang et
al. (2008) have highlighted the effect of the hysteresis of particle phase transitions on the
distribution of solid and aqueous aerosols. The aerosol phase is generally regulated by the
hysteresis loop. Atmospheric particles containing inorganic salts remain solid until the RH
reaches the DRH (deliquescence relative humidity). At the DRH, the solid particle
spontaneously absorbs water to become a saturated aqueous solution. However, the liquid
particle does not crystallize when the RH is below the DRH (Seinfeld and Pandis, 2006).
Therefore, another possible pathway exists to suppress the $N_2O_5$ hydrolysis, i.e., the inorganic
particles might be in solid phase without organic coating. Further studies need to be
conducted to evaluate the hysteresis effect on the $N_2O_5$ hydrolysis and organic coating.
**3.4  Sensitivity studies of organic aerosol hygroscopicity to the nitrate formation**
In section 3.3, the WRF-Chem model considerably improves nitrate simulations when
considering the $N_2O_5$ heterogeneous hydrolysis and organic coating effects. Organic aerosols
(OA) are broadly classified as primary OA (POA) directly emitted and SOA formed in the
atmosphere, some of which are water soluble. In order to explore the effects of different OA
coating on the nitrate formation, additional four sensitivity studies are conducted, in which
half of SOA (C1-case), all SOA (C2-case), all SOA and half of POA (C3-case), and all SOA
and POA (C4-case) are involved in coating, respectively.
Figure 11 shows the Taylor diagram (Taylor, 2001) to present the variance, bias and
correlation of the observed and simulated nitrate concentrations in the four sensitivity cases at
CRAES site during the episode. In the C1-case, when half of SOA is considered to involve in
coating, the simulated nitrate concentration is the best consistent with the observation, with a
correlation coefficient of 0.96. In the C2-case with all SOA assumed to engage in coating, the
correlation coefficient decreases to be 0.95. The normalized standardized deviation (NSD) is
1.02 for the C1-case and C2-case, showing the model overestimation in these two cases. With
half of POA involved in coating, the NSD is very close to 1.0 (0.99), indicating the simulated
nitrate concentration in the C3-case is almost the same as the observation on average, but the
correlation coefficient of 0.94 is less than those in the C1-case and C2-case. When all of OA
is assumed as the coating, the bias between simulated and observed nitrate concentrations is
the largest, and the effect of POA on suppressing the $N_2O_5$ heterogeneous hydrolysis might be
overestimated in the C4-case.
Sensitivity results show that the effects of different organic compounds on suppressing
the $N_2O_5$ heterogeneous hydrolysis to form nitrate varies, depending on the content of water
soluble OA.  Laboratory  and  field  measurements  have  revealed  that  OA  becomes
progressively oxidized and more hygroscopic during the aging process in the atmosphere
(Jimenez et al., 2009). The OA hygroscopicity constitutes a necessary prerequisite for
accurately representing the organic coating effect on the $N_2O_5$ heterogeneous hydrolysis.
According to the simulations in the present study, in BTH, not all of SOA can serve as the
coating to suppress the nitrate formation, and the effect of POA on coating might be
neglected. Xing et al. (2019) have shown that in BTH, the heterogeneous SOA formed by
irreversible uptake of glyoxal and methylglyoxal on wet aerosol surfaces contributes about 30%
of the SOA mass during haze days. Considering the possible heterogeneous SOA contribution
of other carbonyl compounds and the atmospheric aging of OA, about half of SOA should
likely be hygroscopic and involved in coating.

**4    Conclusion**
Nitrate aerosol has constituted a main component of $PM_{2.5}$ with implementation of
aggressive emission control strategies since 2013 in BTH. In the study, the Riemer09
parameterization is implemented into the WRF-Chem model to simulate the nitrate formation
from the $N_2O_5$ heterogeneous hydrolysis referred to as the most important pathway of the
nitrate formation at nighttime. A heavy haze episode from 10 to 27 February 2014 in BTH is
simulated using the WRF-Chem model to verify the effect of organic coating on the $N_2O_5$
heterogeneous hydrolysis and its consequent contribution to the nitrate formation. Analyses
of synoptic fields show a stagnant weather condition with the prevailing southerly wind in the
low-level atmosphere in BTH and surrounding areas during the episode, facilitating
accumulation of air pollutants and heavy haze formation.
The WRF-Chem model performs reasonably in predicting the temporal variations of the
meteorological parameters compared to observations in Beijing. The model generally
reproduces well the temporal variations and spatial distributions of air pollutants against
observations at monitoring sites in BTH. In addition, the simulated diurnal profiles of sulfate,
ammonium and SOA are also in good agreements with the measurements at CRAES site in
Beijing.
The Riemer09 parameterization with all the SOA assumed to be involved in coating
considerably improves the nitrate simulations compared to the measurements at CRAES site
in Beijing. When organic coating is not considered in the Riemer09 parameterization, the
model overestimates the nitrate concentration against the measurements. On average, organic
coating decreases nitrate concentrations by 4.7 $\mu g\ m^{-3}$ or 8.4% in BTH during the episode.
Furthermore, the $N_2O_5$ heterogeneous hydrolysis with organic coating contributes about 30.1%
of nitrate concentrations, and 11.6% of the $PM_{2.5}$ concentration in BTH, playing a
considerable role in the haze formation.
Sensitivity studies reveal that the OA hygroscopicity is a necessary prerequisite for
accurately evaluating the organic coating effect on the $N_2O_5$ heterogeneous hydrolysis. In the
present study, POA might not serve as coating and about half of SOA should be involved in
coating to suppress the nitrate formation. Future studies still need to be conducted to further
predict the OA hygroscopicity, in order to more precisely represent the organic coating effect
on the $N_2O_5$ heterogeneous hydrolysis in chemical transport models.


*Author contribution.* Guohui Li, as the contact author, provided the ideas and financial
support, verified the conclusions, and revised the paper. Lang Liu conducted a research,
designed the experiments, carried the methodology out, performed the simulation, processed
the data, prepared the data visualization, and prepared the manuscript with contributions from
all authors. Jiarui Wu and Xia Li provided the treatment of meteorological data, analyzed the
study data, validated the model performance, and reviewed the manuscript. Suixin Liu, Yang

Qian, Tian Feng, and Jiamao Zhou provided the observation data used in the study, synthesized the observation, and reviewed the paper. Xuexi Tie and Junji Cao provided critical reviews pre-publication stage.

*Acknowledgements*. This work is financially supported by the National Key R&D Plan (Quantitative Relationship and Regulation Principle between Regional Oxidation Capacity of Atmospheric and Air Quality (2017YFC0210000)) and National Research Program for Key Issues in Air Pollution Control.

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

**Table 1** WRF-Chem model configurations.

| Regions | Beijing-Tianjin-Hebei (BTH) |
|---|---|
| Simulation period | February 10 to 27, 2014 |
| Domain size | 200 × 200 |
| Domain center | 38.0°N, 116.0°E |
| Horizontal resolution | 6km × 6km |
| Vertical resolution | 35 vertical levels with a stretched vertical grid with spacing ranging from 30m near the surface, to 500m at 2.5km and 1km above 14km |
| Microphysics scheme | WSM 6-class graupel scheme (Hong and Lim, 2006) |
| Boundary layer scheme | MYJ TKE scheme (Janjić, 2002) |
| Surface layer scheme | MYJ surface scheme (Janjić, 2002) |
| Land-surface scheme | Unified Noah land-surface model (Chen and Dudhia, 2001) |
| Long-wave radiation scheme | Goddard longwave scheme (Chou and Suarez, 2001) |
| Short-wave radiation scheme | Goddard shortwave scheme (Chou and Suarez, 1999) |
| Meteorological boundary and initial conditions | NCEP 1°×1° reanalysis data |
| Chemical initial and boundary conditions | MOZART 6-hour output (Horowitz et al., 2003) |
| Anthropogenic emission inventory | SAPRC-99 chemical mechanism emissions (Zhang et al., 2009) |
| Biogenic emission inventory | MEGAN model developed by Guenther et al. (2006) |
| Model spin-up time | 24 hours |






Table 2 Statistics for model performance.

|  | MFB (%) | MFE (%) | MB | RMSE | IOA |
|---|---|---|---|---|---|
| [a]Temperature | -3.7 | 8.5 | -0.2 ˚C | 1.7 ˚C | 0.94 |
| [a]Relative humidity | 5.2 | 15.7 | 2.6 % | 10.9 % | 0.89 |
| [a]Wind speed | 16.3 | 38.6 | 0.3 m s$^{-1}$ | 1.0 m s$^{-1}$ | 0.73 |
| [a]Wind direction | 26.3 | 54.6 | 21.9 ˚ | 90.4 ˚ | 0.66 |
| [b]PM$_{2.5}$ | -3.0 | 13.5 | -6.3 μg m$^{-3}$ | 27.6 μg m$^{-3}$ | 0.96 |
| [b]O$_3$ | 2.1 | 28.3 | 1.4 μg m$^{-3}$ | 10.3 μg m$^{-3}$ | 0.91 |
| [b]NO$_2$ | 10.6 | 16.2 | 6.6 μg m$^{-3}$ | 13.0 μg m$^{-3}$ | 0.92 |
| [b]SO$_2$ | 6.1 | 23.9 | 7.6 μg m$^{-3}$ | 27.8 μg m$^{-3}$ | 0.85 |
| [b]CO | 10.6 | 18.5 | 0.2 mg m$^{-3}$ | 0.5 mg m$^{-3}$ | 0.90 |
| [c]SOA | -14.5 | 52.9 | -1.2 μg m$^{-3}$ | 15.5 μg m$^{-3}$ | 0.83 |
| [c]Sulfate | 23.8 | 53.0 | 4.5 μg m$^{-3}$ | 26.5 μg m$^{-3}$ | 0.88 |
| [c]Ammonium | 22.2 | 44.2 | 2.9 μg m$^{-3}$ | 16.4 μg m$^{-3}$ | 0.90 |
| [c]Nitrate | 7.1 | 37.1 | 0.1 μg m$^{-3}$ | 19.0 μg m$^{-3}$ | 0.96 |

a, b, and c represent the meteorological parameter averaged over 12 meteorological sites in
Beijing, the air pollutant averaged over all ambient monitoring stations in BTH, and the
aerosol component at the CRAES site in Beijing, respectively.

 **Figure Captions**


Figure 1 WRF-Chem simulation domain with topography. The red filled circles show the
locations of the cities with ambient air quality monitoring sites, and the size of the
circles represents the number of sites in each city. The blue filled rectangle denotes the
CRAES observation site in Beijing.

Figure 2 Distributions of average winds (black flag vectors), geopotential heights (blue lines),
temperature (red lines), and relative humidity (contour fill) at (a) and (b) 500hPa and
(c) and (d) 850hPafrom 10 to 27 February 2014, respectively.

Figure 3 Temporal variations of simulated (red line) and observed (black dots) meteorological
parameters of near-surface (a) temperature, (b) relative humidity, (c) wind speed, and
(d) wind direction averaged at 12 meteorological sites in Beijing from 10 to 27
February 2014.

Figure 4 Relationships between observed and simulated mass concentrations of $PM_{2.5}$, $O_3$,
$NO_2$, $SO_2$, and CO in Beijing, Tianjin, and Hebei from 10 to 27 February 2014. The
red line is the linear regression between observations and simulations, and the black
dashed line presents the 1:1 line.

Figure 5 Comparison of observed (black dots) and simulated (red line) diurnal profiles of
near surface hourly (a) $PM_{2.5}$, (b) $O_3$, (c) $NO_2$, (d) $SO_2$, and (e) CO averaged over all
ambient monitoring stations in BTH from 10 to 27 February 2014.

Figure 6 Spatial distributions of average (a) $PM_{2.5}$, (b) $O_3$, (c) $NO_2$, and (d) $SO_2$ mass
concentrations from 10 to 27 February 2014. Colored dots, colored contour, and black
arrows are observations, simulations, and simulated surface winds, respectively.

Figure 7 Comparison of observed (black dots) and simulated (red line) diurnal profiles of
hourly (a) SOA, (b) sulfate, and (c) ammonium concentrations at CRAES site in
Beijing from 10 to 27 February 2014.

Figure 8 Temporal variations of observed (black dot) and the simulated (Green line: H0-case;
Blue line: C0-case; Red line: B-case) nitrate concentrations at CRAES site in Beijing
from 10 to 27 February 2014.

Figure 9 Spatial distributions of average nitrate contributions of (a) the $N_2O_5$ heterogeneous
hydrolysis and (b) organic coating in BTH from 10 to 27 February 2014.

Figure 10 Temporal variation of the simulated $\gamma_{N_2O_5}$ in the B-case in Beijing from 10 to 27
February 2014.

Figure 11 Taylor diagram (Taylor, 2001) to present the variance, bias and correlation of the
observed and simulated nitrate concentrations at CRAES site in Beijing from 10 to 27
February 2014.






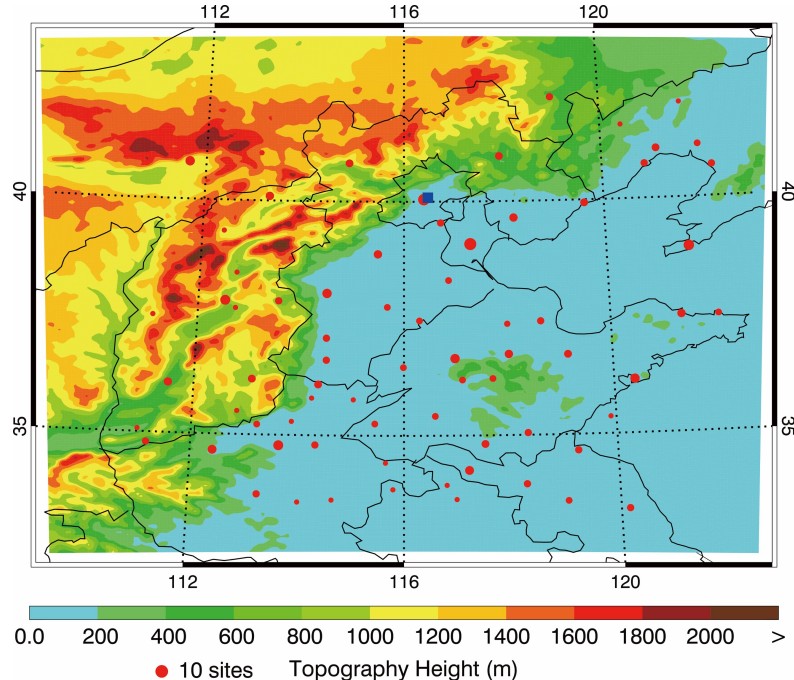



Figure 1 WRF-Chem simulation domain with topography. The red filled circles show the
locations of the cities with ambient air quality monitoring sites, and the size of the circles
represents the number of sites in each city. The blue filled rectangle denotes the CRAES
observation site in Beijing.





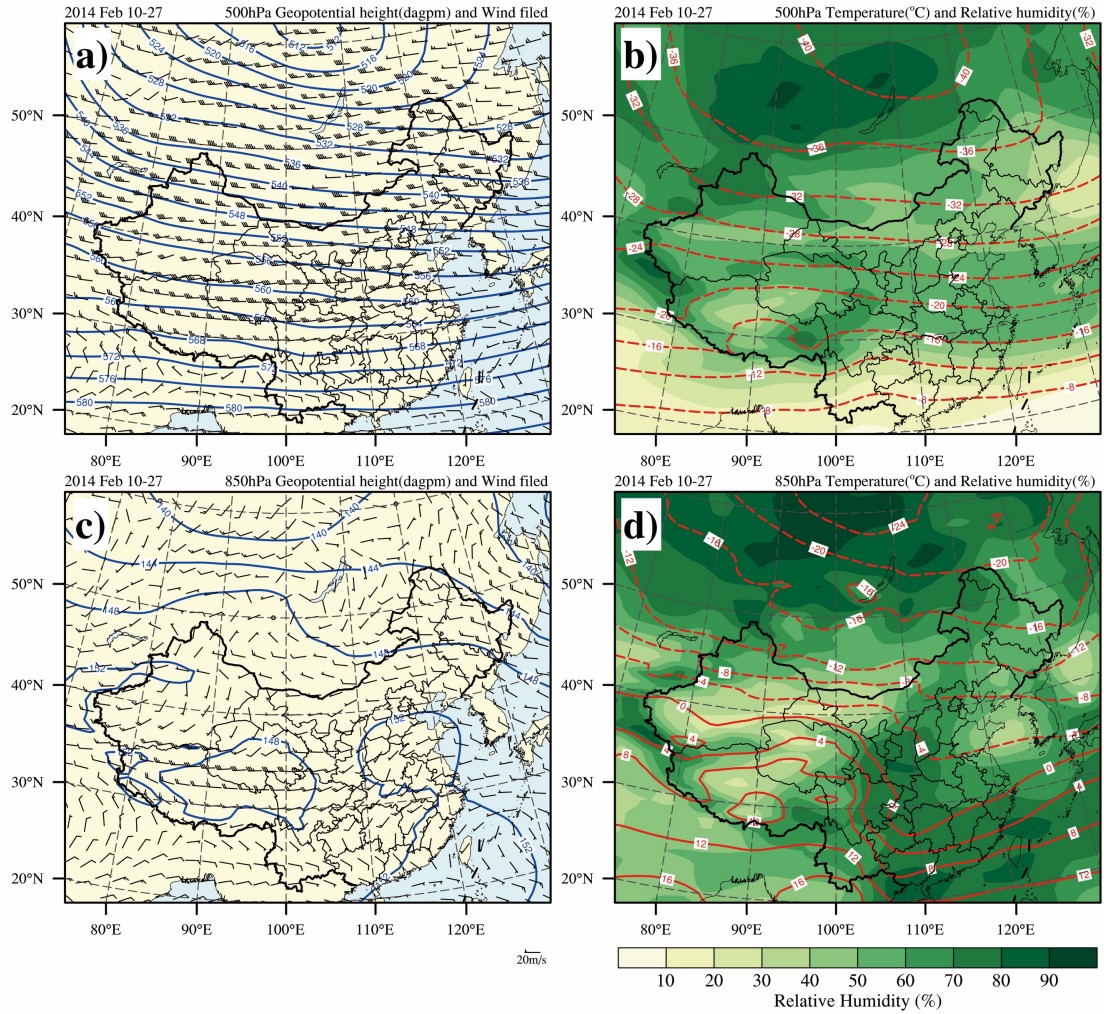



Figure 2 Distributions of average winds (black flag vectors), geopotential heights (blue lines),
temperature (red lines), and relative humidity (contour fill) at (a) and (b) 500hPa and (c) and
(d) 850hPafrom 10 to 27 February 2014, respectively.





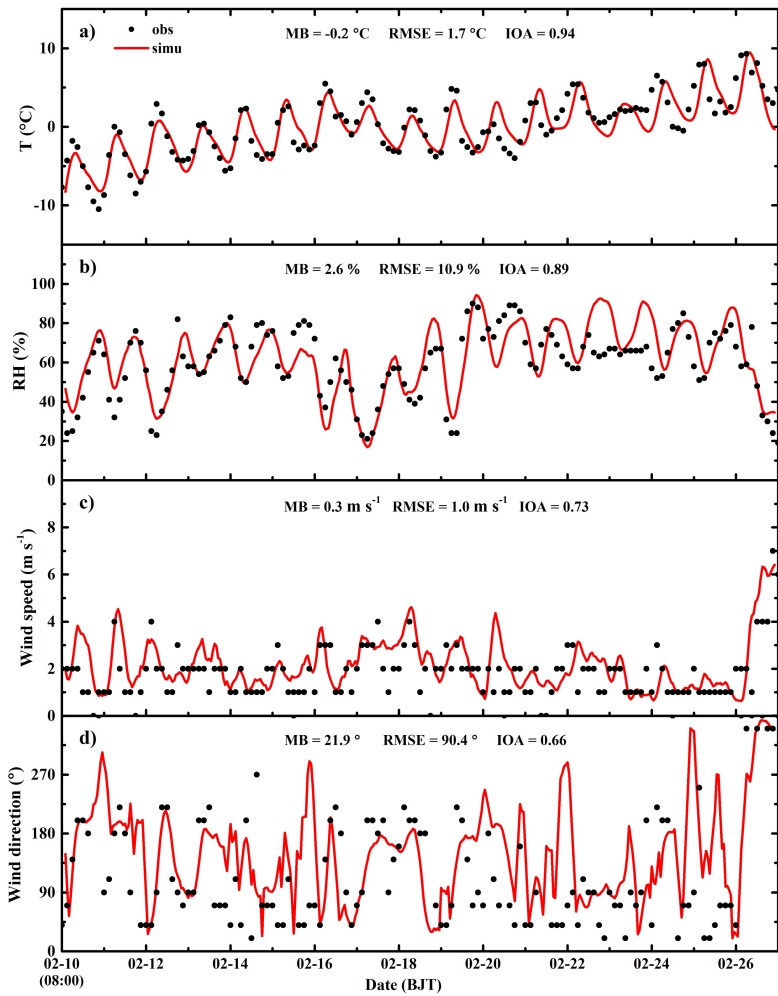

Figure 3 Temporal variations of simulated (red line) and observed (black dots) meteorological
parameters of near-surface (a) temperature, (b) relative humidity, (c) wind speed, and (d)
wind direction averaged at 12 meteorological sites in Beijing from 10 to 27 February 2014.

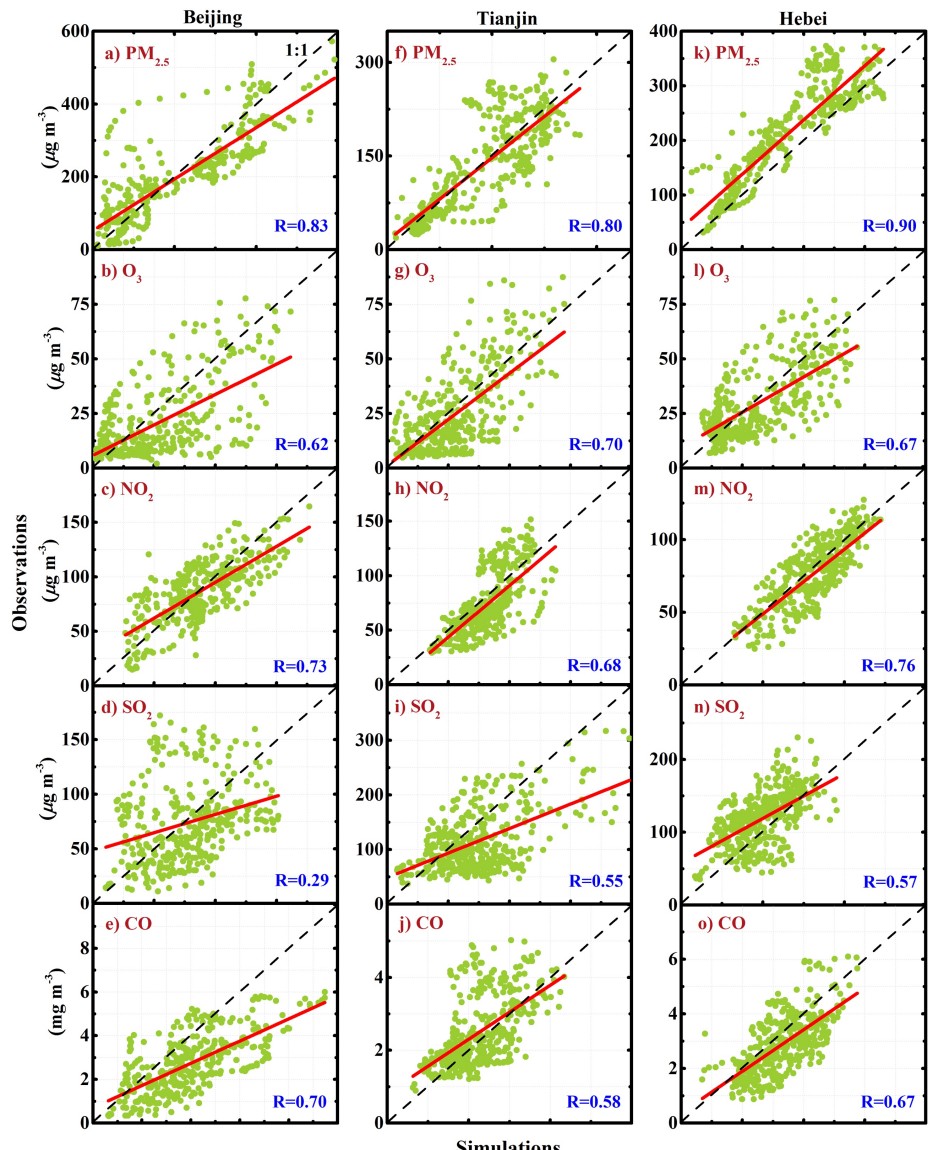

Figure 4 Relationships between observed and simulated mass concentrations of PM$_{2.5}$, O$_3$,
NO$_2$, SO$_2$, and CO in Beijing, Tianjin, and Hebei from 10 to 27 February 2014. The red line
is the linear regression between observations and simulations, and the black dashed line
presents the 1:1 line.

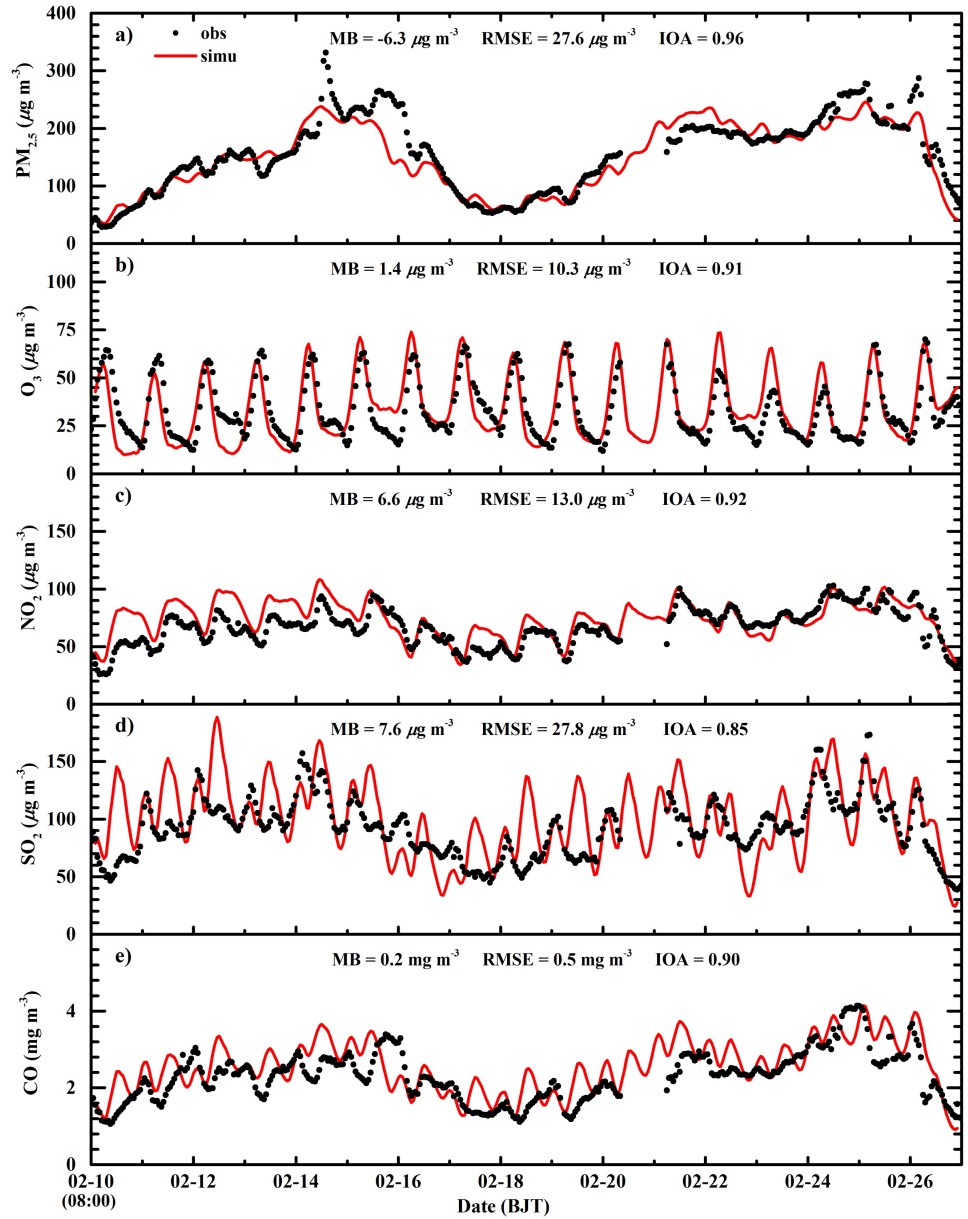

Figure 5 Comparison of observed (black dots) and simulated (red line) diurnal profiles of near surface hourly (a) $PM_{2.5}$, (b) $O_3$, (c) $NO_2$, (d) $SO_2$, and (e) CO averaged over all ambient monitoring stations in BTH from 10 to 27 February 2014.

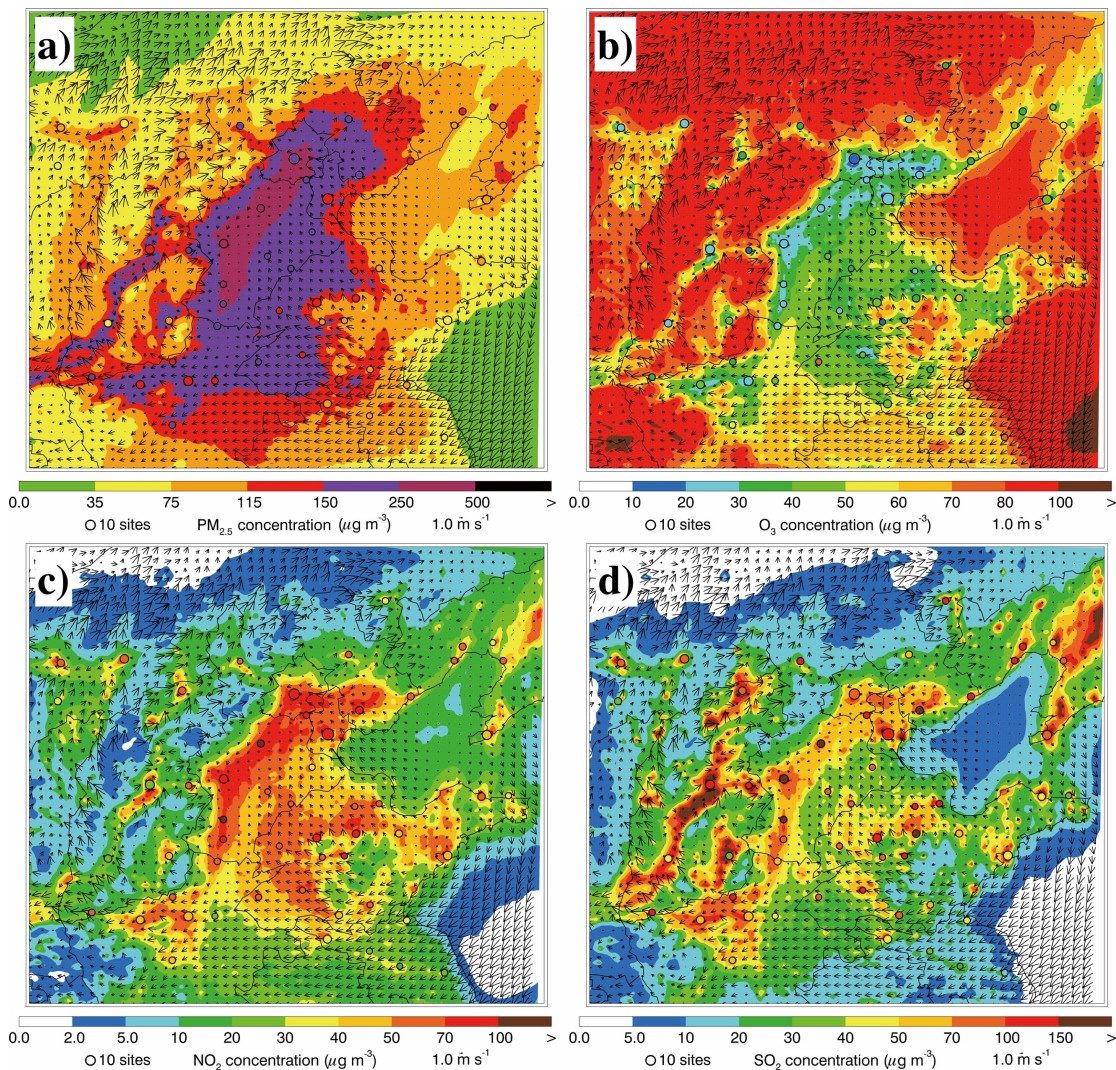

Figure 6 Spatial distributions of average (a) PM$_{2.5}$, (b) O$_3$, (c) NO$_2$, and (d) SO$_2$ mass concentrations from 10 to 27 February 2014. Colored dots, colored contour, and black arrows are observations, simulations, and simulated surface winds, respectively.

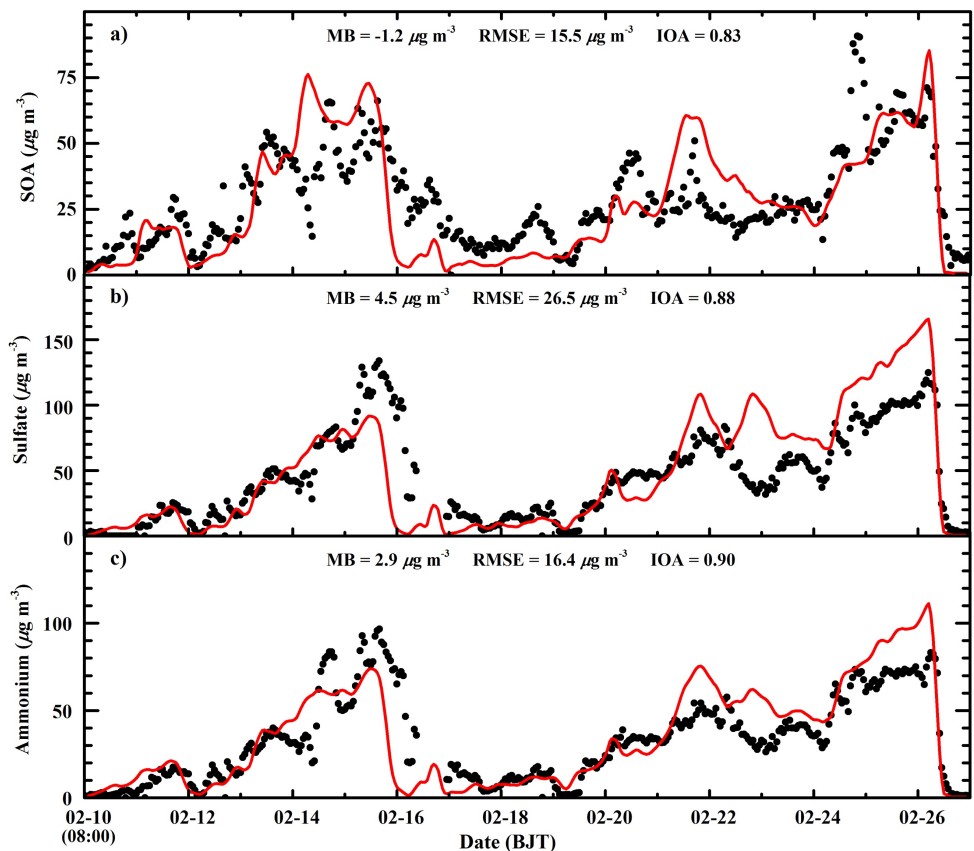

Figure 7 Comparison of observed (black dots) and simulated (red line) diurnal profiles of hourly (a) SOA, (b) sulfate, and (c) ammonium concentrations at CRAES site in Beijing from 10 to 27 February 2014.

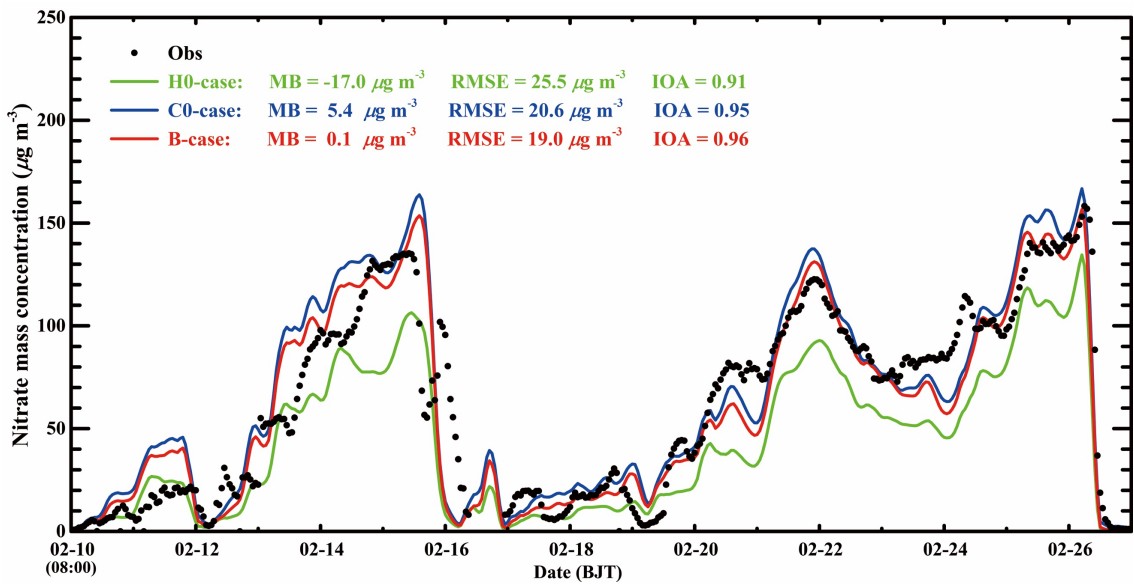

Figure 8 Temporal variations of observed (black dot) and the simulated (Green line: H0-case;
Blue line: C0-case; Red line: B-case) nitrate concentrations at CRAES site in Beijing from 10
to 27 February 2014.

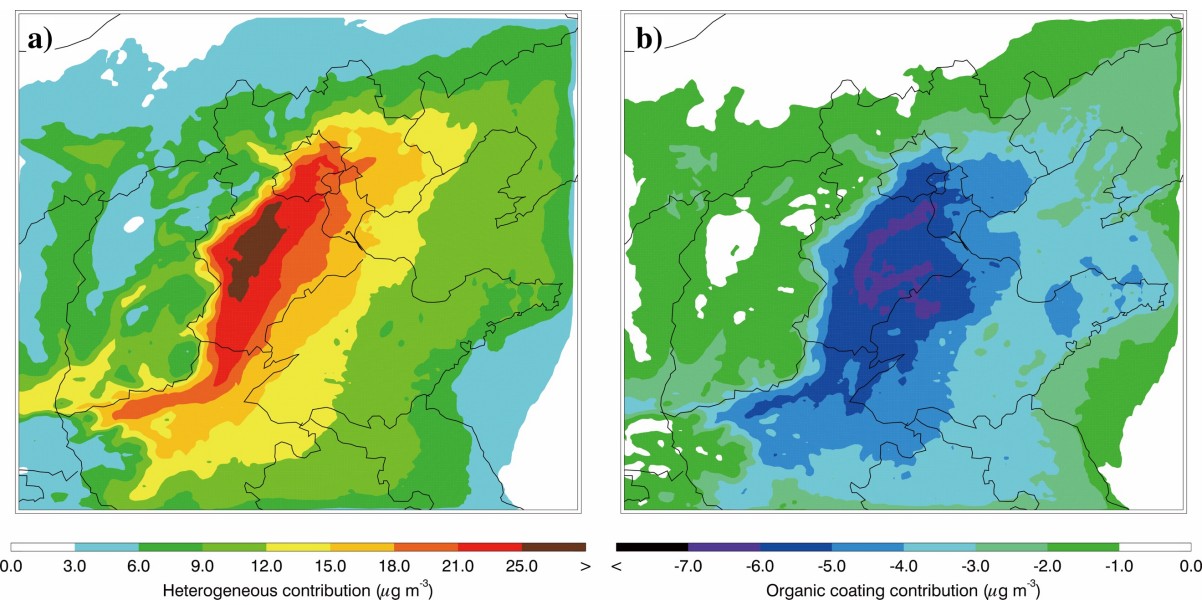

Figure 9 Spatial distributions of average nitrate contributions of (a) the $N_2O_5$ heterogeneous hydrolysis and (b) organic coating in BTH from 10 to 27 February 2014.

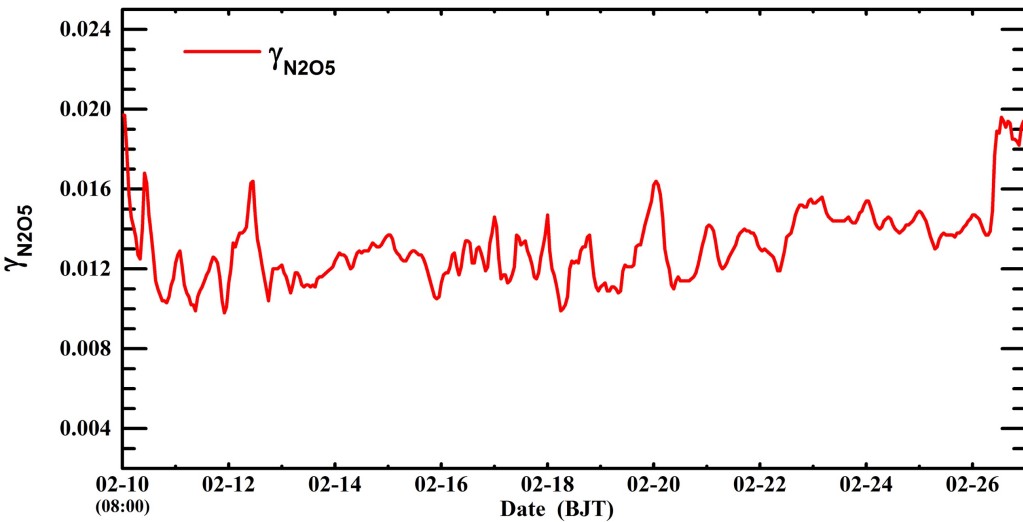

Figure 10 Temporal variation of the simulated $\gamma_{N_2O_5}$ in the B-case in Beijing from 10 to 27 February 2014.

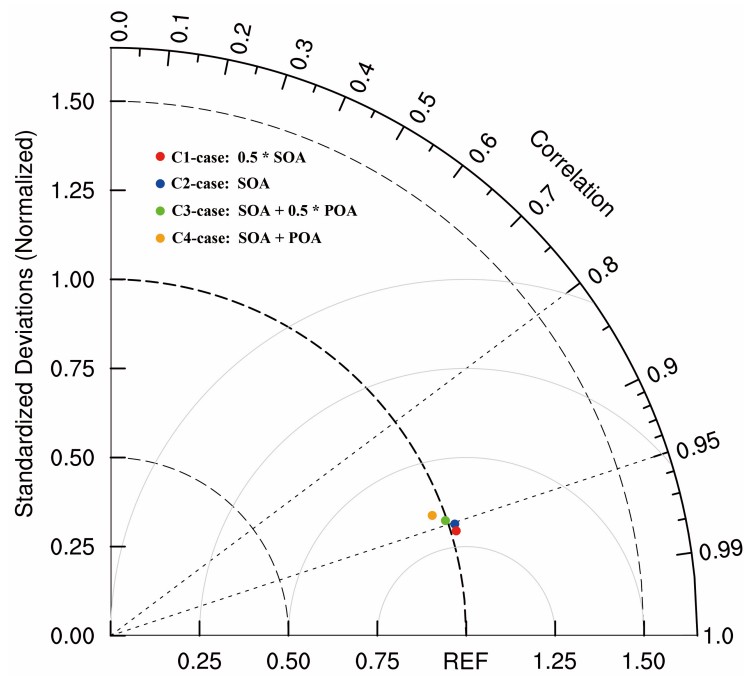

Figure 11 Taylor diagram (Taylor, 2001) to present the variance, bias and correlation of the observed and simulated nitrate concentrations at CRAES site in Beijing from 10 to 27 February 2014.