# Peer review of "Effects of organic coating on the nitrate formation by suppressing the N2O5 heterogeneous hydrolysis: A case study during wintertime in Beijing-Tianjin-Hebei (BTH)"

_Atmospheric Chemistry and Physics, 2019_

## Referee Comment (RC1) · Anonymous Referee #1 · 13 Apr 2019

In this study, the WRF-Chem model was used to understand the effects of organic coating on particles on N2O5 heterogeneous hydrolysis. The study has very good model performance on simulating air pollutants and shows that the coating of organic is important for nitrate formation. But more details are needed to show how the model was improved. As the model performance is really good, other researchers can learn and improve their simulations. Thus, I think a major revision is needed before publication.

Comments:

[Figure]

1. The writing can be improved by correcting unprofessional usages. Some examples in abstract: a. WRF-Chem not WRF-CHEM b. particulate matter not particulate matters c. Line 20, why "referred to as" is needed? d. Lines 22-23, "the" is not needed in front of every noun.

2. The model used is not clear at all to the readers, which make the model results unreliable. For example, the WRF-Chem was based on studies published in around 2010, have the new features of new versions of WRF-Chem been incorporated? It said that the CMAQ aerosol module was used, but what version, AERO5 or AERO6? Isorropia II has been out since 2007, why Isorropia 1.7 is still in use? What is the gas phase mechanism? SOA contributions from glyoxal and methylglyoxal were added, but how? Have the results been validated?

3. The emission is from Zhang et al 2009 (in table 1: SAPRC-99 chemical mechanism emissions), which is a much coarser resolution and have more than 5 years' difference in time (published in 2009 and simulation in 2014). How did the emission is processed before running WRF-Chem? With the large uncertainties in the emission inventories for China, it is hard to believe that the nitrate concentration has MB of 0.1 ugm-3, so as other components. I encourage more details to be added.

4. The method is questionable. a. All inorganic components are assumed as core? This is not reasonable. SOA, nitrate, sulfate, and ammonia are all formed secondarily, it is not right to assume that SOA is the shell while others are the core. Also, primary OA should also lay in the center as the core, right? Why it is not? b. Line 130, why H2O concentration is not considered in the reaction rate? c. Line 137, ðİŻ¿2 is what? d. Lines 144-145, ðİŻ¿N2O5,core is calculated by Eq 4?

5. MFB and MFE are recommended by several studies for PM validation. I would like to see they are used to validate the model results.

6. For the meteorological parameters, there are studies suggesting benchmarks. It is subjective to say "reproduces" and "well consistent with". Similar for air pollutants.

7. How the PM2.5 components were measured? What instruments? Any published results? The SOA performance with MB of -1.2 ugm-3 is really interesting as there is no model so far can capture such high SOA concentrations (>75 ugm-3). This is very important for the authors to show very detailed description of their SOA model.

8. Sulfate is most underpredicted in current models for as high as 150ug/m3. It is also not clear why this model predict even higher compared to observations, as you are using CMAQ module for that.

9. Nitrate results is perfect to has a MB of 0.1 ug/m3. But meteorological conditions have larger certainties, how can this not affect the nitrate performance at all?

10. As the results show that the organic coating is important. It is essential to show that the detailed values of the parameters involving in the processes. For example, the gamma values.
* * *

---

## Referee Comment (RC2) · Anonymous Referee #2 · 19 Apr 2019

This paper presents the implementation of N2O5 hydrolysis on organic-coating particles in WRF-Chem, and analyzes the impact of such implementation on the simulation of nitrate and other aerosol particles. Overall, nitrate concentration is reduced due to this newly-added pathway for N2O5 heteregenous hydrolysis.

While the parameterization for N2O5 hydrolysis on organic-coating particles is not new, it appears that this paper is among the first to add this parameterization into WRF-Chem. I recommend the paper to be published after addressing the following comments.

During the study time period of Feb 10 to Feb 26, there were multiple times that RH values are below 40% (especially during daytime). For sulfate particles at least, their phase is regulated by the hysteresis loop - solid sulfate will not become liquid until RH is above 80% and liquid sulfate particle will not become solid until RH is below 40%. Hence, there is another possible pathway to suppress N2O5 hydrolysis - that is - the inorganic particles can be in solid phase even without organic coating. Authors should at least mention how the particle phases are treated in the model? Are all sulfate particles in aqueous phase? And discuss additionally possible pathway. The following paper is recommended for the discussion. Wang, J., A. A. Hoffmann, R. Park, D. J. Jacob, and S. T. Martin, 2008. Global distribution of solid and aqueous sulfate aerosols: effect of the hysteresis of particle phase transitions, J. Geophys. Res., 113, D11206.

---

## Author Comment (AC1) · 31 May 2019

**Reply to Anonymous Referee #1**

We thank the reviewer for the careful reading of the manuscript and helpful comments. We have revised the manuscript following the suggestion, as described below.

In this study, the WRF-Chem model was used to understand the effects of organic coating on particles on $N_2O_5$ heterogeneous hydrolysis. The study has very good model performance on simulating air pollutants and shows that the coating of organic is important for nitrate formation. But more details are needed to show how the model was improved. As the model performance is really good, other researchers can learn and improve their simulations. Thus, I think a major revision is needed before publication.

**1 Comment:** The writing can be improved by correcting unprofessional usages. Some examples in abstract: a. WRF-Chem not WRF-CHEM; b. particulate matter not particulate matters; c. Line 20, why "referred to as" is needed? d. Lines 22-23, "the" is not needed in front of every noun.

**Response:** a. We have changed "WRF-CHEM" to "WRF-Chem" in the manuscript; b. we have changed "particulate matters" as "particulate matter"; c. we have revised the sentence as "*The $N_2O_5$ heterogeneous hydrolysis is the most important pathway of the nitrate formation at nighttime.*"; and d. we have revised the sentence as "*… to evaluate contributions of the $N_2O_5$ heterogeneous hydrolysis to nitrate formation and effects of organic coating*".

**2 Comment:** The model used is not clear at all to the readers, which make the model results unreliable. For example, the WRF-Chem was based on studies published in around 2010, have the new features of new versions of WRF-Chem been incorporated? It said that the CMAQ aerosol module was used, but what version, AERO5 or AERO6? Isorropia II has been out since 2007, why Isorropia 1.7 is still in use? What is the gas phase mechanism? SOA contributions from glyoxal and methylglyoxal were added, but how? Have the results been validated?

**Response:** We have clarified in Section 2.1: "*A new flexible gas phase chemical module has been developed and implemented into the version of the WRF-Chem model, which can be utilized with different chemical mechanisms, including CBIV, RADM2, and SAPRC. The gas-phase chemistry is solved by an Eulerian backward Gauss-Seidel iterative technique with*"

*a number of iterations, inherited from NCAR-HANK (Hess et al., 2000). In the study, the SAPRC99 chemical mechanism is used based on the available emission inventory. For the aerosol simulations, the CMAQ/models3 aerosol module (AERO5) developed by US EPA has incorporated into the model (Binkowski and Roselle, 2003)."*

We have also clarified in Section 2.1:

" *ISORROPIA (version 1.7) is used to predict the thermodynamic equilibrium between the ammonia-sulfate-nitrate-chloride-water aerosols and their gas phase precursors of H2SO4-HNO3-NH3-HCl-water vapor. It is worth noting that the most recent extension of ISORROPIA, known as ISORROPIA II, has incorporated a larger number aerosol species (Ca, Mn, K salts) and is designed to be a superset of ISORROPIA (Fountoukis and Nenes, 2007). Considering that crustal species are not considered in the study, ISORROPIA (version 1.7) is still used to calculate inorganic components and ISORROPIA II is imperative to be incorporated into the WRF-Chem model in future studies. In addition, a parameterization of sulfate heterogeneous formation involving aerosol liquid water (ALW) has been developed and implemented into the model, which has successfully reproduced the observed rapid sulfate formation during haze days (Li et al., 2017). The sulfate heterogeneous formation from $SO_2$ is parameterized as a first order irreversible uptake by ALW surfaces, with a reactive uptake coefficient of $0.5 \times 10^{-4}$ assuming that there is enough alkalinity to maintain the high iron-catalyzed reaction rate.*

*The OA module is based on the VBS approach with aging and detailed information can be found in Li et al. (2011b). The POA components from traffic-related combustion and biomass burning are represented by nine surrogate species with saturation concentrations ($C^*$) ranging from $10^{-2}$ to $10^6$ $\mu g$ $m^{-3}$ at room temperature (Shrivastava et al., 2008), and assumed to be semi-volatile and photochemically reactive (Robinson et al., 2007). The SOA formation from each anthropogenic or biogenic precursor is calculated using four semi-volatile VOCs with effective saturation concentrations of 1, 10, 100, and 1000 $\mu g$ $m^{-3}$ at 298 K. The SOA formation via the heterogeneous reaction of glyoxal and methylglyoxal is parameterized as a first-order irreversible uptake by aerosol particles with an uptake coefficient of $3.7 \times 10^{-3}$ (Liggio et al., 2005; Zhao et al., 2006; Volkamer et al., 2007). The OA module has reasonably reproduced the POA and SOA concentration against measurements, and detailed model performance can be found in Li et al. (2011b), Feng et al. (2016), and Xing et al. (2019).*"

**3 Comment:** The emission is from Zhang et al. 2009 (in table 1: SAPRC-99 chemical mechanism emissions), which is a much coarser resolution and have more than 5 years' difference in time (published in 2009 and simulation in 2014). How did the emission is processed before running WRF-Chem? With the large uncertainties in the emission inventories for China, it is hard to believe that the nitrate concentration has MB of 0.1 μg m$^{-3}$, so as other components. I encourage more details to be added.

**Response:** We have clarified in Section 2.1: "*The anthropogenic emission inventory with a horizontal resolution of 6 km is developed by Zhang et al. (2009), with the base year of 2013, including industry, transportation, power plant, residential and agriculture sources. The Model of Emissions of Gases and Aerosols from Nature (MEGAN) is used to calculate the biogenic emissions online (Guenther et al., 2006).*". About the model performance, please refer to **9 Comment**.

**4 Comment:** The method is questionable. a. All inorganic components are assumed as core? This is not reasonable. SOA, nitrate, sulfate, and ammonia are all formed secondarily, it is not right to assume that SOA is the shell while others are the core. Also, primary OA should also lay in the center as the core, right? Why it is not? b. Line 130, why H$_2$O concentration is not considered in the reaction rate? c. Line 137, $\gamma_2$ is what? d. Lines 144-145, $\gamma_{N_2O_5,core}$ is calculated by Eq 4?

**Response:** a. We have clarified in Section 2.2: "*In this study, the parameterization of $\gamma_{N_2O_5}$ follows Riemer03 and Riemer09. In the parameterization, the primary emission compounds such as element carbon, insoluble organic matter (most part of POA), insoluble inorganic and mineral dust particles are assumed to serve as a nucleus of aerosols. Condensation of soluble chemical components and further water vapor on the surface of the nucleus forms an aqueous layer. The nucleus and the aqueous layer are assumed as unified "core" (aqueous core) in Riemer03 and Riemer09 parameterizations. In Riemer03 parameterization, soluble inorganic components including sulfate and nitrate are taken into consideration for suppressing the N$_2$O$_5$ heterogeneous hydrolysis uptake in the aqueous core, and the parameterization of $\gamma_{N_2O_5}$ is defined as:*"

$$\gamma_{N_2O_5} = f \cdot \gamma_1 + (1-f) \cdot \gamma_2 \qquad\qquad (Eq.\ 4)$$

*with $\gamma_1 = 0.02$ and $\gamma_2 = 0.002$, and f is defined as:*

$$f = \frac{m_{SO_4^{2-}}}{m_{SO_4^{2-}} + m_{NO_3^-}} \qquad\qquad (Eq.\ 5)$$

$m_{SO_4^{2-}}$ and $m_{NO_3^-}$ are the aerosol mass concentrations of soluble sulfate and nitrate.

In Riemer09 parameterization, unreactive organic layers are further considered for the suppression of $N_2O_5$ hydrolysis by covering the aqueous core. Organic layers may be formed by secondary organic aerosols, and such layers may consist of a single layer of molecules (monolayered coatings) or of several molecule layers (multilayered coatings) on the surface of the aqueous core. These organic layers are assumed as organic "coating" (shell) in the Riemer09 parameterization. The resistor scheme to calculate $\gamma_{N_2O_5}$ in Riemer09 parameterization is parameterized as follows:

$$\frac{1}{\gamma_{N_2O_5}} = \frac{1}{\gamma_{N_2O_5,core}} + \frac{1}{\gamma_{N_2O_5,coat}} \qquad (Eq.\ 6)$$

where $\gamma_{N_2O_5,core}$ is the reaction probability of the aqueous core which is calculated using Eq. 4".

b. Aerosol liquid water contribution is included in the calculation of the available aerosol surface area density ($S$).

c. $\gamma_2$ is a constant in the Riemer03 parameterization, we have revised the Eq.4 as: "$\gamma_{N_2O_5} = f \cdot \gamma_1 + (1-f) \cdot \gamma_2$".

d. The $\gamma_{N_2O_5,core}$ is calculated by Eq. 4.

**5 Comment:** MFB and MFE are recommended by several studies for PM validation. I would like to see they are used to validate the model results.

**Response:** We have clarified in Section 2.3:

" In this study, the mean bias (MB), root mean square error (RMSE), the index of agreement (IOA), mean fractional bias (MFB) and mean fractional error (MFE) are used to evaluate the model performance in simulating air pollutants.

$$MB = \frac{1}{N}\sum_{i=1}^{N}(P_i - O_i) \qquad (Eq.\ 11)$$

$$RMSE = \left[\frac{1}{N}\sum_{i=1}^{N}(P_i - O_i)^2\right]^{\frac{1}{2}} \qquad (Eq.\ 12)$$

$$IOA = 1 - \frac{\sum_{i=1}^{N}(P_i - O_i)^2}{\sum_{i=1}^{N}(|P_i - \bar{O}| + |O_i - \bar{O}|)^2} \qquad (Eq.\ 13)$$

$$MFB = \frac{1}{N}\sum_{i=1}^{N}\frac{P_i - O_i}{(P_i + O_i)/2} \qquad (Eq.\ 14)$$

$$MFE = \frac{1}{N}\sum_{i=1}^{N}\frac{|P_i-O_i|}{(P_i+O_i)/2} \qquad \text{(Eq. 15)''}$$

Additionally, we have clarified in Section 3.2:

" *Furthermore, the MFB and MFE between simulations and observations are also calculated to evaluate the model performance in simulating meteorological parameters and air pollutants (Table 2). Boylan and Russell (2006) have proposed that MFB should be within ±60% and MFE should be below 75% for a satisfactory model performance. For the simulation in the B-case, MFB values are within 27% and MFE values are below 55%, indicating that the model performance is satisfactory.*"

**6 Comment:** For the meteorological parameters, there are studies suggesting benchmarks. It is subjective to say "reproduces" and "well consistent with". Similar for air pollutants.

**Response:** we have clarified in Section 3.2:

" *Furthermore, the MFB and MFE between simulations and observations are also calculated to evaluate the model performance in simulating meteorological parameters and air pollutants (Table 2). Boylan and Russell (2006) have proposed that MFB should be within ±60% and MFE should be below 75% for a satisfactory model performance. For the simulation in the B-case, MFB values are within 27% and MFE values are below 55%, indicating that the model performance is satisfactory.*"

**7 Comment:** How the PM$_{2.5}$ components were measured? What instruments? Any published results? The SOA performance with MB of -1.2 µg m$^{-3}$ is really interesting as there is no model so far can capture such high SOA concentrations (>75 µg m$^{-3}$). This is very important for the authors to show very detailed description of their SOA model.

**Response:** We have clarified in Section 2.4: "*Additionally, hourly OC and EC concentrations are measured using a thermal/optical reflectance carbon analyzer (OCEC RT-4, Sunset Lab, USA) at Chinese Research Academy of Environmental Sciences (CRAES, 40.04°N, 116.40°E) in Beijing (Liu et al., 2018; Wei et al., 2014). Hourly sulfate, nitrate, ammonium, and other inorganic ions are sampled and analyzed by ion chromatography (URG 9000S, Thermo Fisher Scientific, USA).*

*The OC/EC ratio approach is used to derive the SOA mass concentration from EC and OC measurements as follows (Strader, 1999; Cao et al., 2004):*

$$POC = EC \times \left(\frac{POC}{EC}\right) \qquad \text{(Eq. 16)}$$

$$SOC = OC - POC \qquad \text{(Eq. 17)}$$

$$SOA = SOC \times \left(\frac{SOA}{SOC}\right) \qquad \text{(Eq. 18)}$$

*Where POC and SOC are the primary OC and secondary OC, respectively. In the present study, $\frac{POC}{EC}$ and $\frac{SOA}{SOC}$ are assumed to be 2.4 and 1.6, respectively, based on the previous studies (Cao et al., 2007; Aiken et al., 2008; Yu et al., 2009) and detailed information about the approach can be found in Feng et al. (2016). It is worth noting that those assumed $\frac{POC}{EC}$ and $\frac{SOA}{SOC}$ could potentially affect the model-measurement comparisons.*"

We have also clarified in Section 2.1:

" *The OA module is based on the VBS approach with aging and detailed information can be found in Li et al. (2011). The POA components from traffic-related combustion and biomass burning are represented by nine surrogate species with saturation concentrations (C\*) ranging from $10^{-2}$ to $10^6$ µg m$^{-3}$ at room temperature (Shrivastava et al., 2008), and assumed to be semi-volatile and photochemically reactive (Robinson et al., 2007). The SOA formation from each anthropogenic or biogenic precursor is calculated using four semi-volatile VOCs with effective saturation concentrations of 1, 10, 100, and 1000 µg m$^{-3}$ at 298 K. The SOA formation via the heterogeneous reaction of glyoxal and methylglyoxal is parameterized as a first-order irreversible uptake by aerosol particles with an uptake coefficient of $3.7 \times 10^{-3}$ (Liggio et al., 2005; Zhao et al., 2006; Volkamer et al., 2007). The OA module has reasonably reproduced the POA and SOA concentration against measurements, and detailed model performance can be found in Li et al. (2011b), Feng et al. (2016), and Xing et al. (2019).*"

**8 Comment:** Sulfate is most underpredicted in current models for as high as 150 µg m$^{-3}$. It is also not clear why this model predicts even higher compared to observations, as you are using CMAQ module for that.

**Response:** We have clarified in Section 2.1: "*In addition, a parameterization of sulfate heterogeneous formation involving aerosol liquid water (ALW) has been developed and implemented into the model, which has successfully reproduced the observed rapid sulfate*

*formation during haze days (Li et al., 2017). The sulfate heterogeneous formation from SO₂ is parameterized as a first order irreversible uptake by ALW surfaces, with a reactive uptake coefficient of 0.5×10⁻⁴ assuming that there is enough alkalinity to maintain the high iron-catalyzed reaction rate.*"

**9 Comment:** Nitrate results are perfect to has a MB of 0.1 µg m⁻³. But meteorological conditions have larger certainties, how can this not affect the nitrate performance at all?

**Response:** We have clarified in Section 3.3: "*It is worth noting that the MB for nitrate aerosols at CRAES site in the B-case is close to zero, but the RMSE is still rather large, reaching 19.0 µg m⁻³, showing considerable underestimation and overestimation, caused by uncertainties of meteorological fiends and emissions. For example, the model overestimates nitrate concentrations on 11, 13, and 14 February and underestimation on 24 February against measurements. In addition, the early occurrence of intensified winds in the morning on 16 February in simulations cause rapid falloff of nitrate concentrations, leading to substantial model biases.*"

**10 Comment:** As the results show that the organic coating is important. It is essential to show that the detailed values of the parameters involving in the processes. For example, the gamma values.

[revised manuscript text omitted]

---

## Author Comment (AC2) · 31 May 2019

**Reply to Anonymous Referee #2**

We thank the reviewer for the careful reading of the manuscript and helpful comments. We have revised the manuscript following the suggestion, as described below.

This paper presents the implementation of $N_2O_5$ hydrolysis on organic-coating particles in WRF-Chem, and analyzes the impact of such implementation on the simulation of nitrate and other aerosol particles. Overall, nitrate concentration is reduced due to this newly-added pathway for $N_2O_5$ heterogeneous hydrolysis.

While the parameterization for $N_2O_5$ hydrolysis on organic-coating particles is not new, it appears that this paper is among the first to add this parameterization into WRF- Chem. I recommend the paper to be published after addressing the following comments.

**Comments:** During the study time period of Feb 10 to Feb 26, there were multiple times that RH values are below 40% (especially during daytime). For sulfate particles at least, their phase is regulated by the hysteresis loop - solid sulfate will not become liquid until RH is above 80% and liquid sulfate particle will not become solid until RH is below 40%. Hence, there is another possible pathway to suppress $N_2O_5$ hydrolysis - that is - the inorganic particles can be in solid phase even without organic coating. Authors should at least mention how the particle phases are treated in the model? Are all sulfate particles in aqueous phase? And discuss additionally possible pathway. The following paper is recommended for the discussion. Wang, J., A. A. Hoffmann, R. Park, D. J. Jacob, and S. T. Martin, 2008. Global distribution of solid and aqueous sulfate aerosols: effect of the hysteresis of particle phase transitions, J. Geophys. Res., 113, D11206.

**Response:** We have clarified in Section 3.3: "*It is worth noting that, in the study, the assumption of metastable aerosols is used or the water soluble aerosol is assumed to be only in liquid state in simulations. However, Wang et al. (2008) have highlighted the effect of the hysteresis of particle phase transitions on the distribution of solid and aqueous aerosols. The aerosol phase is generally regulated by the hysteresis loop. Atmospheric particles containing inorganic salts remain solid until the RH reaches the DRH (deliquescence relative humidity). At the DRH, the solid particle spontaneously absorbs water to become a saturated aqueous solution. However, the liquid particle does not crystallize when the RH is below the DRH (Seinfeld and Pandis, 2006). Therefore, another possible pathway exists to suppress the $N_2O_5$*

*hydrolysis, i.e., the inorganic particles might be in solid phase without organic coating. Further studies need to be conducted to evaluate the hysteresis effect on the $N_2O_5$ hydrolysis and organic coating.*"

**References**

Wang, J., Hoffmann, A. A., Park, R. J., Jacob, D. J., and Martin, S. T.: Global distribution of solid and aqueous sulfate aerosols: Effect of the hysteresis of particle phase transitions, J. Geophys. Res.-Atmos., 113, D11206, https://doi.org/10.1029/2007JD009367, 2008.

Seinfeld, J. H. and Pandis, S. N.: Atmospheric Chemistry and Physics: From Air Pollution to Climate Change, 2nd Edn., John Wiley & Sons Inc., New York, 2006.